# Occurrence of Capnophilic Lactic Fermentation in the Hyperthermophilic Anaerobic Bacterium *Thermotoga* sp. Strain RQ7

**DOI:** 10.3390/ijms231912049

**Published:** 2022-10-10

**Authors:** Nunzia Esercizio, Mariamichela Lanzilli, Simone Landi, Lucio Caso, Zhaohui Xu, Genoveffa Nuzzo, Carmela Gallo, Emiliano Manzo, Sergio Esposito, Angelo Fontana, Giuliana d’Ippolito

**Affiliations:** 1Institute of Biomolecular Chemistry (ICB), National Research Council (CNR), Via Campi Flegrei 34, 80078 Pozzuoli, Italy; 2Department of Biology, University of Naples “Federico II”, Via Cinthia, 80126 Napoli, Italy; 3Department of Biological Sciences, Bowling Green State University, Bowling Green, OH 43403, USA

**Keywords:** hydrogen, lactic acid, CO_2_ valorization, ATPase, bio-based process, green chemistry, *Thermotoga neapolitana*, *Thermotoga maritima*

## Abstract

Capnophilic lactic fermentation (CLF) is an anaplerotic pathway exclusively identified in the anaerobic hyperthermophilic bacterium *Thermotoga neapolitana*, a member of the order Thermotogales. The CO_2_-activated pathway enables non-competitive synthesis of hydrogen and L-lactic acid at high yields, making it an economically attractive process for bioenergy production. In this work, we discovered and characterized CLF in *Thermotoga* sp. strain *RQ7*, a naturally competent strain, opening a new avenue for molecular investigation of the pathway. Evaluation of the fermentation products and expression analyses of key CLF-genes by RT-PCR revealed similar CLF-phenotypes between *T. neapolitana* and *T.* sp. strain RQ7, which were absent in the non-CLF-performing strain *T. maritima*. Key CLF enzymes, such as PFOR, HYD, LDH, RNF, and NFN, are up-regulated in the two CLF strains. Another important finding is the up-regulation of V-ATPase, which couples ATP hydrolysis to proton transport across the membranes, in the two CLF-performing strains. The fact that V-ATPase is absent in *T. maritima* suggested that this enzyme plays a key role in maintaining the necessary proton gradient to support high demand of reducing equivalents for simultaneous hydrogen and lactic acid synthesis in CLF.

## 1. Introduction

The order Thermotogales represents a group of anaerobic hyperthermophilic bacteria that are recognized as promising microorganisms for bio-hydrogen production from sugars via dark fermentation [1,2,3,4]. A number of *Thermotoga* and *Pseudothermotoga* species demonstrated efficient fermentation performance using different substrates, including simple monosaccharides (hexoses and pentoses) and complex polysaccharides (e.g., starch, lactose, sucrose, and cellobiose) [1,4,5,6,7]. The ability of these microorganisms to generate green hydrogen from organic wastes, such as food scrapes, lignocellulosic biomasses and glycerol, further strengthens their values in the production of sustainable bioenergy [4].

Theoretically, up to 4 mole of hydrogen can be produced from each mole of hexose, with acetic acid and CO_2_ as the byproducts [8], as shown in Equation (1).
C_6_H_12_O_6_ + 2H_2_O → 2 CH_3_COOH + 4H_2_ + 2CO_2_(1)

In practice, more reduced volatile fatty acids, such as lactic acid and ethanol, accumulate during dark fermentation, which competes for reducing power and lowers the experimental yield of hydrogen [9].

Recently, we reported a new pathway named capnophilic lactic fermentation (CLF) in *Thermotoga neapolitana* (Figure 1) [10]. Under CO_2_ atmosphere, CLF enables the non-competitive synthesis of L-lactic acid and hydrogen at the same time. Besides a classic catabolic branch from sugars to acetate, the pathway also has an anabolic branch from acetyl-CoA to lactate, enabled by sequential actions of pyruvate:ferredoxin oxidoreductase (PFOR) and lactate dehydrogenase (LDH) [10,11,12,13,14]. To support the concomitant production of hydrogen and lactic acid, the additional NADH requirement is likely satisfied by a CO_2_-induced metabolic process involving NAD-Ferredoxin oxidoreductase (RNF) and NADH-dependent reduced ferredoxin:NADP oxidoreductase (NFN) [9,14,15,16].

Enzymatic reductive carboxylation of acetate to lactate offers a biological mechanism to convert CO_2_ into commodity chemicals and thus has a considerable biotechnological significance [10,17]. To improve the fixation rate of CO_2_ into lactate, a strategy to increase acetate uptake has been attempted in *T. neapolitana* by heterologous expression of *Thermus thermophilus* acetyl-CoA synthetase (ACS), which is involved in irreversible acetate assimilation [15]. Genetic engineering of *T. neapolitana* remains challenging, however, due to the lack of appropriate selective markers and low efficiencies of the transformation methods available to this organism, such as liposome-mediated transformation and electroporation [15,18].

*T.* sp. strain RQ7, isolated from marine sediments of Ribeira Quente (Azores), is the first *Thermotoga* strain to be found naturally competent, making genetic manipulation of *Thermotoga* more feasible [19,20]. The construction of *Thermotoga-E. coli* shuttle vectors and the development of new selective schemes based on the pyrEF system have greatly expanded the toolbox for genetic manipulation of *Thermotoga* spp. [15,19,20,21]. The genomes of *T. neapolitana* and *T.* sp. strain RQ7 share an average nucleotide identity of 98.49%, suggesting that *T.* sp. strain RQ7 is a strain of *T. neapolitana* [22].

The aim of the present work was to assess whether *T.* sp. strain RQ7 was able to perform CLF. Under CO_2_ insufflation to induce CLF, the fermentation parameters of RQ7 were tested, including growth rate, glucose consumption, hydrogen, and organic acid production. The results were compared with the CLF-performing strain *T. neapolitana* and non-CLF-performing strain *T. maritima*. The gene expression levels of CLF key enzymes were also investigated to highlight the molecular differences of the three strains.

## 2. Results

### 2.1. Fermentation Products under N_2_ and CO_2_ in T. neapolitana, T. sp. Strain RQ7, and T. maritima

*T. neapolitana* (*Tnea*), *T.* sp. strain RQ7 (*Trq7*), and *T. maritima* (*Tmar*) were grown on glucose under N_2_ and CO_2_ atmospheres in static batches to assess the fermentation performance after 24 and 48 h. As shown in Figure 2, in *Tnea* and *Trq7*, CO_2_ accelerated the consumption of glucose, which was almost depleted within 48 h (~87%). In *Tmar*, the glucose consumption was similar under N_2_ and CO_2_ conditions. The consumption rates were significantly lower in *Tmar* than the other two strains_._

For hydrogen production, all strains exhibited no significant differences between N_2_ and CO_2_ sparging after either 24 or 48 h (Figure 3). Total hydrogen production was lower in *Tmar* compared to the other two strains, parallel to the lower consumption of glucose.

Acetic acid and lactic acid are the main organic end products in sugar fermentation by *Thermotogales* [3,8,23]. Interestingly, CO_2_ did not affect acetate production in any strains compared to N_2_ sparging, at either 24 or 48 h (Figure 4a). In *Tnea* and *Trq7*, CO_2_ insufflation doubled the lactate level at both 24 and 48 h, whereas in *Tmar,* no significant difference in lactate levels was observed at either time point (Figure 4b).

CLF is characterized by increased lactate synthesis without affecting hydrogen production, which was confirmed in *Tnea* and *Trq7*. Approximately 2.6 mol hydrogen/mol glucose was observed in *Tnea* and *Trq7* regardless of the sparging method, whereas the lactic acid yield roughly increased by 0.2 mol/mol glucose by replacing N_2_ with CO_2_ (from 0.27 to 0.47 mol/mol glucose) (Table 1). The ratio between lactic acid and acetic acid was nearly doubled in both *Tnea* and *Trq7* after sparging with CO_2_. These results demonstrated the activation of the CLF pathway in *Tnea* and *Trq7*, which manifested that *Trq7* is a CLF-performing strain. No activation of CLF was observed in *Tmar* since the lactate levels remained low in both sparging cases.

### 2.2. Gene Expression Level of Key CLF Enzymes

As reported in Figure 1, the CLF pathway involves glycolysis followed by at least four transformations requiring (1) PFOR, the reversible enzyme responsible for both the catabolic reaction from pyruvate to acetyl-CoA and the anabolic reaction from acetyl-CoA and CO_2_ to pyruvate; (2) LDH, for the synthesis of lactic acid from pyruvate; (3) HYD, containing the H-cluster for the reduction of H^+^ to hydrogen; (4) phosphate acetyltransferase (PTA) and acetate kinase (ACK), reversible enzymes participating in the acetate dissimilation pathway from acetyl-CoA to acetate and vice-versa. In order to underline the molecular differences among the three strains, we compared the gene expression levels of these five enzymes after 24 h of fermentation under the N_2_ and CO_2_ sparging (Figure 5).

The qRT-PCR analysis of the selected genes showed similar transcription levels in *Tnea* and *Trq7*, with 6.2 and 3.2-fold increases in *HYDa*, 19.4 and 8.6-fold increases in *PFOR*, 19.6 and 12.4-fold increases in *LDH*, and 5.45 and 5.14-fold increases in *PTA* (Figure 5). *ACK* did not change significantly under the two sparing treatments in both strains. In *Tmar*, CO_2_ insufflation induced no significant changes in *LDH*, *ACK,* and *PTA*, and only a slight decrease in *HYDa* and a 2.7-fold increase in *PFOR*.

### 2.3. Expression of *Energy-Related Enzymes*

The above results of *Tnea* were in good agreement with our previous data obtained from a transcriptomic study [14], which had also revealed the effects on a number of energy-related enzymes, including flavin-based oxidoreductase enzymes such as NAD-ferredoxin oxidoreductase (*RNF*) (CTN_RS02165), NADH-dependent reduced ferredoxin:NADP oxidoreductase (*NFN*) (CTN_RS04020), and ATPases involved in ion translocation. For this reason, we also analyzed these genes in the three strains under both sparging conditions.

*RFN* and *NFN* transfer electrons between ferredoxin and NAD(P)H and generate an ion gradient across the cell membrane. Sparging with CO_2_ versus N_2_ increased the expression of *NFN* 3.2-fold in *Tnea*, 3.2-fold in *Trq7*, and 1.9-fold in *Tmar*. Meanwhile, *RNF* exhibited a 13.2-fold increase in *Tnea*, 7.7-fold increase in *Trq7*, and no change in *Tmar* (Figure 6).

Ion-translocating ATPases are essential cellular energy converters, which transduce the chemical energy of ATP hydrolysis into transmembrane ionic electrochemical potential gradients [24,25,26]. Gene clusters encoding F- and V-ATPases have been described in *Tnea*, whereas *Tmar* has only F-ATPase gene clusters [27]. In order to determine occurrence of ATPases in Trq7, we mined its genome using the sequences of *Tnea* V-ATPase (CTN_RS04525-CTNRS04550) and *Tnea* F-ATPase (CTN_RS04140- CTN_RS04180) (Table 2). Our results confirmed the absence of V-ATPase subunits in *Tmar* and showed the presence of both F- and V-type ATPases in *Trq7*. As expected, *Tnea* and *Trq7* demonstrated sequence similarities for both V- and F-ATPases with 98–100% of identities for any analyzed subunits.

The bioinformatic results were confirmed by PCR amplification of the catalytic subunits of the F-type (CTN_RS04145) and V-type (CTN_RS04540). *Tnea* and *Trq7* presented both subunits while *Tmar* showed the F-ATPase subunit only (Appendix A).

A further bioinformatic analysis on other species of the Thermotogaceae family, including the genera *Thermotoga* (*T. naphtophila* and *T. petrophila*) and *Pseudothermotoga* (*P. lettingae*, *P. hypogea*, and *P. thermarum*), indicated that these species possessed the F-ATPase gene cluster but not the V-ATPase one (Appendix A). *P. hypogea* showed the presence of an ATPase classified as the V/A-type, with a weak similarity (25–50%) to the V-ATPase complex of *Tnea*, thus representing a different enzyme. On the other hand, the F-ATPases of *Tmar*, *T. naphthophila* and *T. petrophila* showed high levels of identity in all the subunits (around 99–100% with each other), compared to 76–86% of identities to the F-ATPase of *Tnea*.

These results suggested that the V-ATPase gene family is a specific and unique feature of the two CLF-performing strains, *Tnea* and *Trq7*, among *Thermotogales*. Therefore, we compared the expression levels of the ATPases in *Tnea*, *Trq7*, and *Tmar*, after 24 h of fermentation with N_2_ and CO_2_-sparing (Figure 7). For the V-ATPase, we selected the catalytic subunit α(CTN_RS04540) and the regulatory subunit β(CTN_RS04545), and for the F-ATPase, we chose the regulatory subunit α(CTN_RS04145). Both analyzed genes coding for V-ATPase were up-regulated in *Tnea* and *Trq7* under the CLF conditions. These strains, respectively, showed an increase in expression 1.73 and 2.19-fold for subunit α and 4.46 and 2.24-fold for subunitβ (Figure 7). Considering the absence of the V-ATPase, *Tmar* was investigated only for the expression of the F-ATPase. In all three strains, a significant down-regulation of the F-ATPase subunit α was observed, with 1.8, 2.6, and 3.3-fold changes for *Tnea*, *Trq7*, and *Tmar*, respectively (Figure 7).

## 3. Discussion

Capnophilic lactic fermentation (CLF) is a unique pathway of *T. neapolitana*, which is triggered by CO_2_ and induces an increase in lactate whereas it has no impact on the hydrogen yield [10,11,13,14]. The CLF phenotype can be readily recognized by comparing the yields of fermentation products under N_2_ and CO_2_ insufflation [10]. Screening of eight species of the genus *Thermotoga* (*T. neapolitana*, *T. neapolitana* subsp. *capnolactica*, *T. maritima*, *T. naphtophila*, *T. petrophila*, *T. caldifontis*, *T. hypogea*, *T. profunda)* and four species of the genus *Pseudothermotoga* (*P. elfii*, *P. lettingae*, *P. subterranea*, *P. thermarum*) revealed that CLF pathway is only retained in *T. neapolitana* [28]. Moreover, *T. neapolitana* subsp. *capnolactica*, a strain adapted in our laboratory under saturating concentrations of CO_2_, showed a further improved capability of performing CLF in comparison to its parent strain [13].

The analysis of fermentation parameters in *T.* sp. strain RQ7 revealed a clear CLF-phenotype resembling *T. neapolitana* in producing the same hydrogen yield and double lactate under CO_2_. Furthermore, the two strains showed a similar molecular response leading to up-regulation of the genes responsible for hydrogen and lactic acid synthesis with acetate assimilation. It is worth noting that the strong increase in PTA expression well correlated to the acetate assimilation (the ascending pathway in Figure 1), as demonstrated in *T. neapolitana* by both the incorporation of ^13^C-acetate into lactic acid [10,11,13,14] and the moderate increase in lactic acid synthesis due to expression of heterologous ACS [15]. The yields of fermentation products are very different in non-CLF-performing strain *T. maritima*, with no significant differences observed under N_2_ and CO_2_ insufflation. Since *T*. sp. strain RQ7 is a naturally competent strain, genetic manipulation of this organism is more feasible than other *Thermotoga* strains [19,20]. The identification of *T*. sp. strain RQ7 as a CLF-performing strain opens the door to elucidate the molecular mechanism of CLF pathway and to discover new performing strains.

The genomes of *T. neapolitana* and *T.* sp. strain *RQ7* share an average nucleotide identity of 98.49%, making RQ7 a sister strain of *T. neapolitana* [22]. Both bacteria can redirect the flux of NADH and ferredoxin to sustain simultaneous NADH-dependent reactions, thus enabling the CLF pathway. Analyses with qRT-PCR suggested an up-regulation of the key enzymes involved in the anabolic pathway from acetate and CO_2_ to lactic acid in *T. neapolitana* and *T*. sp. strain RQ7, particularly PFOR and LDH. In fact, considering the bi-function of PFOR as both a catabolic (from pyruvate to acetyl-CoA) and an anabolic (from acetyl-CoA and CO_2_ to pyruvate) node, the up-regulation is in good agreement with both the general increase in the metabolism rate induced by CO_2_ and the enhanced demand of pyruvate to feed the downstream synthesis of lactic acid by LDH. The same enzymes are not differentially expressed in *T. maritima* under CO_2_, where the flux from acetyl-CoA to lactic acid remained unchanged. CO_2_ possibly induced a re-organization of cellular redox balance to sustain simultaneous reduction reactions, such as lactate formation from pyruvate and the conversion of protons to molecular hydrogen. A pool of additional reducing equivalents may derive from the flavin-based oxidoreductase enzymes NFN and RNF, which control the supply of reduced ferredoxin and NADH and allow energy conservation based on sodium translocation through the cell membrane. Both enzymes are up-regulated under CO_2_ in *T. neapolitana* and *T*. sp. strain RQ7 but not in *T. maritima*, supporting the hypothesis of their involvement in CLF activation to create a cyclic process for regeneration of NADH and NADPH [14].

Another important aspect of CLF pathway is the generation of ion gradients (Na^+^ and/or H^+^) across the cell membrane. The biochemical characterization of the RNF-ATP synthase supercomplex in *T. maritima* demonstrates that RNF is a Na^+^-dependent ion-translocating respiratory enzyme and is associated with an ATP-synthase activity in the respiratory chain via electrochemical Na^+^ potential [29]. Membrane-bound ATPases are essential in energy conservation, i.e., catalyzing the synthesis or hydrolysis of ATP driven by H^+^/Na^+^ gradients [25,26,30].

ATPases couple the translocation of protons or sodium ions across the membrane to the concomitant synthesis or hydrolysis of ATP [25]. ATPases are grouped into three classes: A-type, F-type, and V-type. A- and F-types are reversible enzymes that harness the energy of an ion gradient across the membrane to synthetize ATP [30,31,32,33]. V-type ATPases can couple the free energy of ATP hydrolysis to proton or sodium translocation and thereby generate an ion-motive force, which is useful for a variety of secondary channel-mediated or active transport [30]. V-type ATPase or its genes have been reported in hyperthermophilic archaea [34,35], while most ATPases in bacteria are F-type. However, it has been documented that the thermophilic bacteria *Thermus thermophilus* [36] and *Clostridium fervidus* [37] have V-type ATPases, although F-type ATPases are found in other *Thermus* species [38]. Our bioinformatic analysis within the Thermotogaceae family revealed that all the analyzed species of *Thermotoga* and *Pseudothermotoga*, including *T. maritima,* have F-type ATPases, whereas *T. neapolitana* and *T*. sp. strain RQ7 are the only ones that possess V-type ATPase. Co-existence of V- and F-type ATPases in *T. neapolitana* and the mere presence of the F-type in *T. maritima* have been reported by Iida et al., 2002 [27]. In this context, the presence and regulation of both F- and V-ATPases in *T. neapolitana* and *T*. sp. strain RQ7 might be crucial to the CLF mechanism. Gene expression analysis of V-ATPases in *T. neapolitana* and *T*. sp. strain RQ7 revealed an up-regulation of this enzyme induced by CO_2_, whereas F-ATPase showed a general down-regulation both in CLF performing and non-performing strains. H^+^/Na^+^ gradients could play an important role in this scheme. The prediction and categorization of ATPase in Na^+^ or H^+^-dependent enzymes can be obtained by sequence analysis of the membrane embedded c/K-oligomers ring which consists a common set of amino acids involved in sodium or proton binding [25]. Based on the sequence analysis of the c/K subunits, the F-ATPases in both *T. maritima* and *T. neapolitana* have been predicted to translocate Na^+^ ions, and the V-ATPase of *T. neapolitana* is supposed to be H^+^-dependent [25]. The Na^+^-dependence of F-ATPase has been demonstrated in *T. maritima* with a connection to the Na^+^-translocating RNF complex [29,39]. Under a CO_2_ atmosphere, the up-regulation of RNF complex and the down-regulation of F-ATPase seem to indicate that a sodium gradient has been generated in the cell by the RNF complex to generate surplus of NADH, which is not exploited by F-ATPase to synthetize ATP. General acceleration of central carbon metabolism observed under CLF forces the cell to dissipate ATP rather than to synthetize it. In *T. neapolitana* and *T*. sp. strain RQ7, the presence and up-regulation of V-ATPase during CLF may be due to the requirement to hydrolyze ATP and “push” cytosolic H^+^ across the membrane against their electrochemical gradient. This generates an ion-motive force (IMF), useful for a variety of secondary channel-mediated or active transport to move substrates or ions over the cellular membrane [30]. V-ATPase based mechanism should be a fundamental step to construct a respiratory chain which contributes to energy conservation and supports redox reactions during CLF.

Parallel studies of the metabolism in *T. neapolitana* and *T*. sp. strain RQ7 are in progress. The preliminary results corroborate the results here presented and will be the subject of a future report on the analogies between the two bacteria.

## 4. Materials and Methods

### 4.1. Biological Materials

Three *Thermotoga* strains were used in this study: *Thermotoga* sp. strain RQ7 (*Trq7*), *Thermotoga maritima* (*Tmar*), and *Thermotoga neapolitana* (*Tnea*) subsp. *capnolactica* (DSM 33003), which were derived from the DSMZ 4359T strain after evolving in our laboratory under saturating concentration of CO_2_ for several years [13]. *Trq7* was gifted by Dr. Harald Huber at the University of Regensburg, Germany. *Tmar* was obtained from DSMZ (DSM 3109). Bacterial cells were anaerobically grown in a modified ATCC 1977 culture medium containing 10 mL/L of filter-sterilized vitamins and trace element solutions (DSM medium 141) together with 10 g/L NaCl, 0.1 g/L KCl, 0.2 g/L MgCl_2_·6H_2_O, 1 g/L NH_4_Cl, 0.3 g/L K_2_HPO_4_, 0.3 g/L KH_2_PO_4_, 0.1 g/L CaCl_2_·2H_2_O, 0.5 g/L cysteine-HCl, 2 g/L yeast extract, 2 g/L tryptone, 5 g/L glucose, 0.001 g/L resazurin (redox indicator) [40].

### 4.2. Bacterial Growth

Aliquots of medium were distributed into 120 mL serum bottles. Pre-cultures (30 mL) were incubated overnight at 80 °C without shaking and used to inoculate (6% *v*/*v*) the samples. Standard culture medium was distributed into 120 mL serum bottles using 30 mL working volume. Oxygen was removed by heating reactors while sparging its content with a stream of pure N_2_ (control) or CO_2_ (trigger of CLF) until the solution was colorless. All the experiments were conducted in triplicates. pH was monitored and adjusted to approximately 7.5 by 1 M NaOH. Sparging followed by pH adjustment was repeated every 24 h. Inoculated bottles were maintained in a heater (Binder ED720) at 80 °C. Cell growth was determined by optical density (OD) at 540 nm (UV/Vis Spectrophotometer DU 730, Beckman Coulter). Aliquots of 2 mL of medium were collected from each sample after 0 h, 24 h and 48 h, centrifuged at 16,000× *g* for 15 min (Hermle Z3236K), and kept at −20 °C until further analyses.

### 4.3. Gas Analyses

Gas (H_2_ and CO_2_) measurements were performed using gas chromatography (GC) on an instrument (Focus GC, Thermo fisher, Waltham MA, USA) equipped with a thermoconductivity detector (TCD) and fitted with a 3 m molecular sieve column (Hayesep Q). N_2_ was used as carrier gas. Analyses were carried out at 24 h and 48 h prior to each gas sparging.

### 4.4. Chemical Analyses

Glucose concentration was determined by the dinitrosalicylic acid method calibrated on a standard solution of 2 g/L glucose [41]. Organic acids were measured by ERETIC 1H NMR as described by Nuzzo et al. [17]. All experiments were performed on a Bruker DRX 600 spectrometer equipped with an inverse TCI CryoProbe. Peak integration, ERETIC measurements and spectrum calibration were obtained by the specific subroutines of the Bruker Top-Spin 3.1 program. Spectra were acquired with the following parameters:  flip angle = 90°, recycle delay = 20 s, SW = 3000 Hz, SI = 16 K, NS = 16, and RG = 1. An exponential multiplication (EM) function was applied to the FID for line broadening of 1 Hz. No baseline correction was used.

### 4.5. RNA Extraction and Real-Time PCR

Aliquots of 20 mL of cultured cells were collected after 24 h from *Tnea, Trq7,* and *Tmar*. Additional samples were collected after 48 h for *Tmar*. Total RNA was extracted using the standard RNA extraction method with TRIzol (Invitrogen, Carlsbad, CA, USA), and cDNA synthesis were performed using the Quantitech^®^ RNA reverse transcription kit (Quiagen, Hilden, Germany). The RNA amounts were measured with a NanoDrop ND-1000 spectrophotometer (Thermo fisher, Waltham, MA, USA). Gene expression analysis was carried out by qRT-PCR. Triplicate quantitative assays were performed using a StepONE plus Real-time PCR system (Applied Biosystems, Foster City, CA, USA) and Platinum SYBR Green qPCR SuperMix (Life Technologies, Carlsbad, CA, USA), with the following program: 5 min at 95 °C, 15 s at 95 °C, 30 s at 60 °C, 40 cycles. The gene of 16S RNA served as the endogenous reference [42]. Calculation of gene expression was carried out using the 2^−ΔΔCt^ method as in Livak and Schmittgen [43]. For each sample, the mRNA levels of selected genes were calculated relative to the calibrator sample for corresponding genes. Primers used for genes expression analyses are listed in Appendix A. 

### 4.6. Bioinformatic Analysis

*Thermotogales* genomes and the protein sequences of *Tnea* V-ATPase and F-ATPase were obtained from the Ensembl bacteria database (https://bacteria.ensembl.org/index.html) (accessed on January 2020). Sequence comparison was performed using the Ensembl BLASTP and NCBI BLASTP tools (https://blast.ncbi.nlm.nih.gov/Blast.cgi, accessed on 1 April 2022).

### 4.7. DNA Extraction and PCR Amplification of ATPase

Genomic DNA was extracted from 20 mL of cultures of *Tnea*, *Trq7* and *Tmar* after 24 h of growth, using a DNA extraction kit (Macherey-nagel, Oensingen, Switzerland) and following the manufacturer’s instructions. For amplification of V-ATPase and F-ATPase, primers were designed on catalytic subunits coded by CTN_RS04540 and CTN_RS04145 (Appendix A). PCR was performed using the following conditions: 5 min at 95 °C, 45 s at 95 °C, 30 s at 60 °C, 72 °C 2 min, 72 °C 10 min, 40 cycles.

### 4.8. Statistics

Each experiment was performed with at least three replicates. Values were expressed as mean ± standard deviation (SD). The statistical significance of comparison between the different treatments (N_2_ vs CO_2_) and species (*Tnea*, *Trq7* and *Tmar*) was calculated through analysis of variance (ANOVA) for hydrogen and organic acids yields (α = 0.05). Differences between means were evaluated for significance using the Tukey–Kramer test. For OD_540_, glucose consumption, and qRT-PCR, the statistical significance of comparison between the N_2_ and CO_2_ was calculated through Student’s *t*-test (*p* ≤ 0.05).

## Figures and Tables

**Figure 1 ijms-23-12049-f001:**
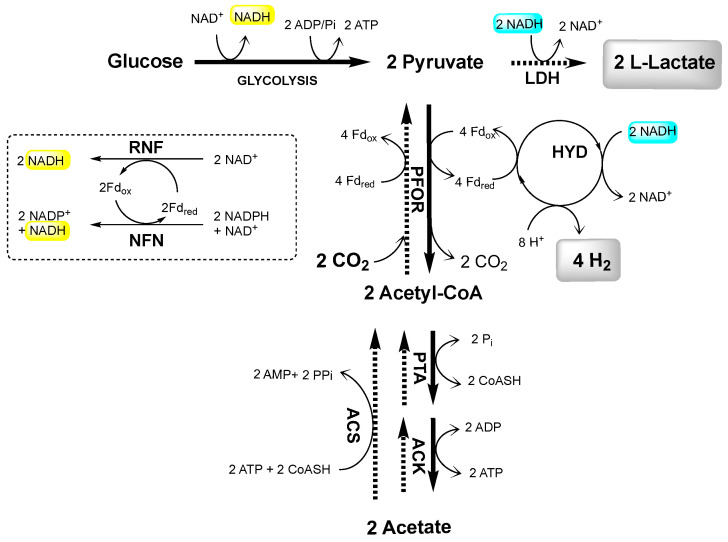
Schematic proposal of the CLF pathway in *T. neapolitana.* ACK, acetate kinase; PTA, phosphotransacetylase; ACS, acetyl-CoA synthetase; PFOR, pyruvate:ferredoxin oxidoreductase; LDH, lactate dehydrogenase; HYD, hydrogenase; RNF, NAD-ferredoxin oxidoreductase; NFN, NADH-dependent reduced ferredoxin:NADP oxidoreductase; Fd, ferredoxin. NADH-source reactions in yellow; NADH-consuming reactions in turquoise. Protons are omitted in the REDOX reactions.

**Figure 2 ijms-23-12049-f002:**
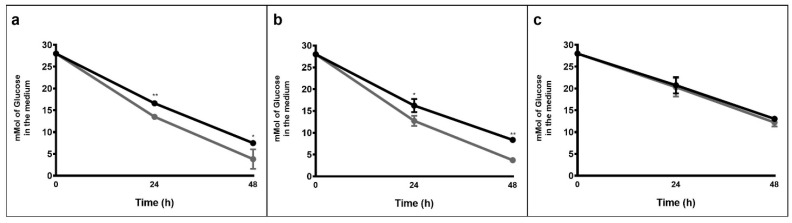
Glucose consumption of *Tnea* (**a**), *Trq7* (**b**), and *Tmar* (**c**) after 24 and 48 h, under N_2_ (black bars) and CO_2_ (gray bars) conditions. Asterisks indicate significant differences: *, *p* ≤ 0.05; **, *p* ≤ 0.001.

**Figure 3 ijms-23-12049-f003:**
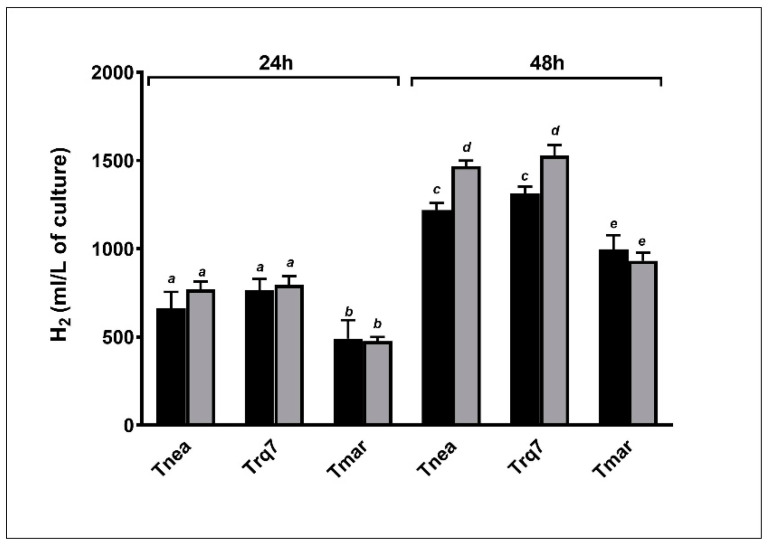
Hydrogen cumulative production after 24 and 48 h, with N_2_ (black bars) and CO_2_ (gray bars). Letters indicate ANOVA significance between treatments and species.

**Figure 4 ijms-23-12049-f004:**
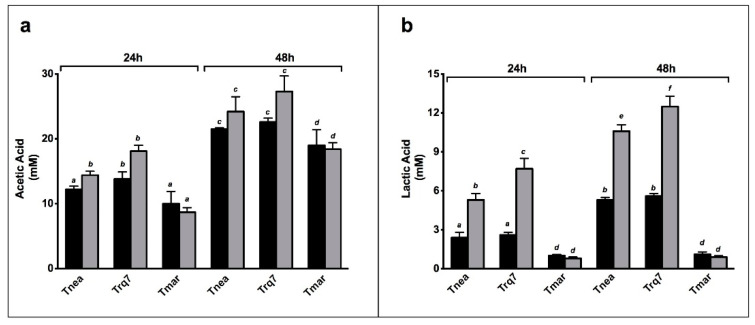
Acetic acid (**a**) and lactic acid (**b**) production under N_2_ (black bars) and CO_2_ (gray bars) conditions, at 24 and 48 h. Letters indicate ANOVA significances between treatments and species.

**Figure 5 ijms-23-12049-f005:**
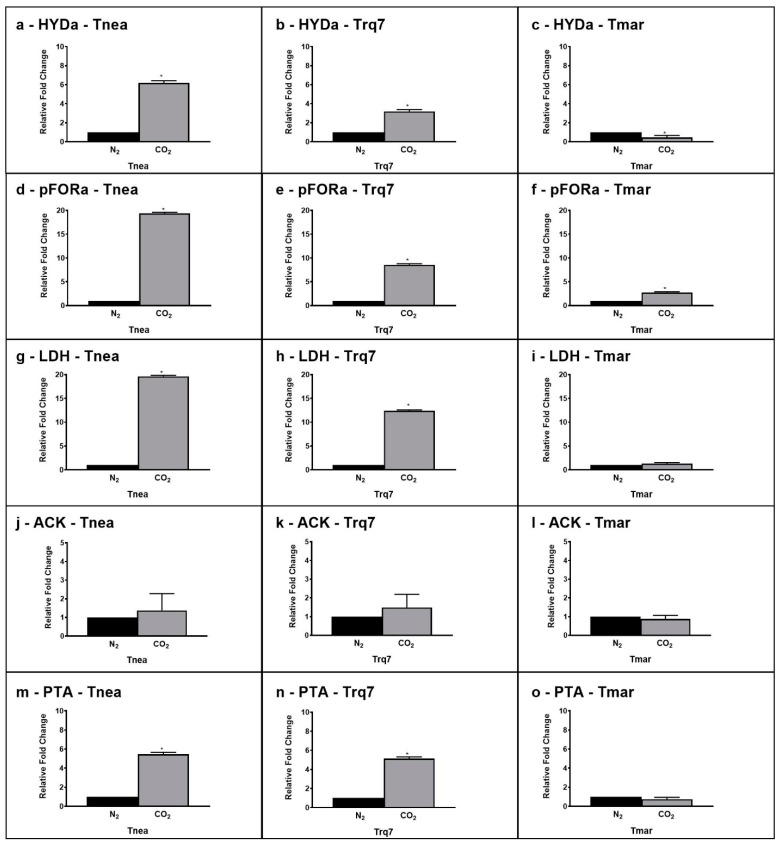
Gene expression levels of HYD (CTN_RS05285 (**a**–**c**)), PFOR (CTN_RS03385 (**d**–**f**)), LDH (CTN_RS03950 (**g**–**i**)), ACK (CTN_RS02020 (**j**–**l**)), and PTA (CTN_RS07210 (**m**–**o**)) after 24 h under N_2_ (black bars) and CO_2_ (gray bars) conditions in *Tnea, Trq7* and *Tmar*. Variations are indicated as relative fold changes in CO_2_ with respect to N_2_. mRNA levels were calculated relative to the expression of 16S RNA. Asterisks indicate significantly different values at *p* ≤ 0.05 (*).

**Figure 6 ijms-23-12049-f006:**
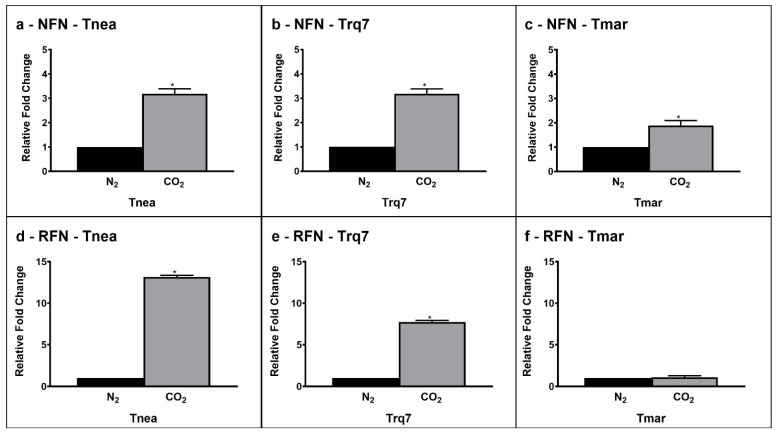
Gene expression levels of NFN (CTN_RS04020 (**a**–**c**)) and RFN (CTN_RS02165 (**d**–**f**)) under N_2_ (black bars) and CO_2_ (gray bars) conditions *Tnea, Trq7* and *Tmar*. Variations are indicated as relative fold changes in CO_2_ with respect to N_2_. mRNA levels were calculated relative to the expression of 16S RNA. Asterisks indicate significantly different values at *p* ≤ 0.05 (*).

**Figure 7 ijms-23-12049-f007:**
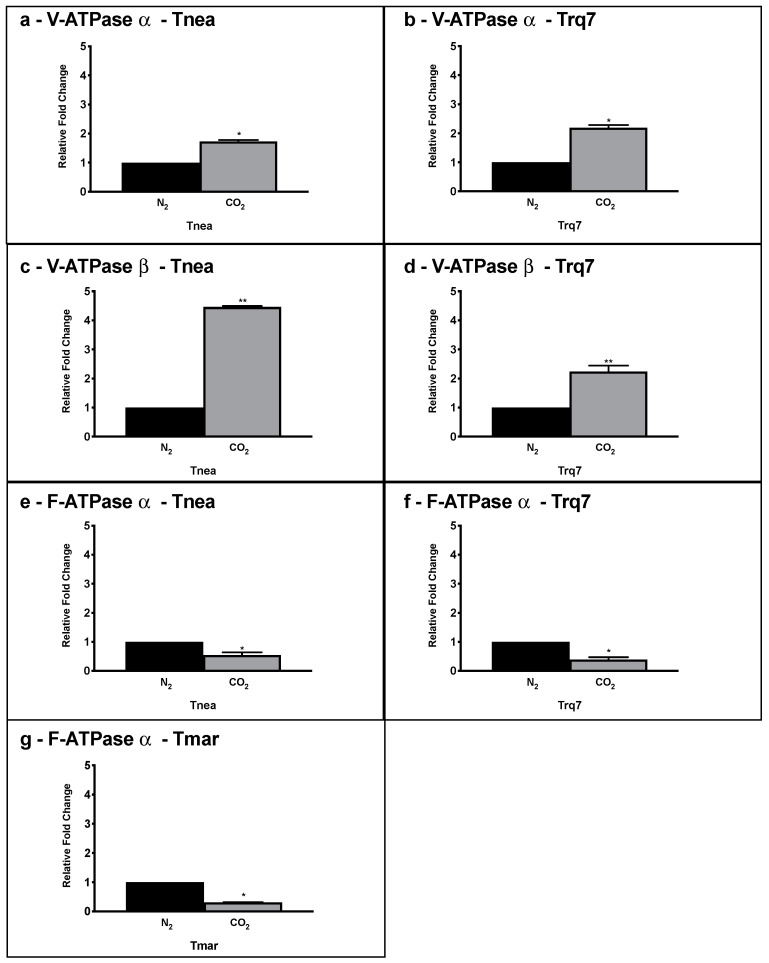
Gene expression levels of V-ATPase subunitα (**a**,**b**) V-ATPase subunit β (**c**,**d**), and F-ATPase subunitα (**e**–**g**), in *Tnea*, *Trq7*, and *Tmar* under N_2_ (black bars) and CO_2_ (gray bars) conditions, measured by qRT-PCR. Variations are indicated as relative fold changes in CO_2_ with respect to N_2_. mRNA levels were calculated relatively to the expression of 16S RNA. Asterisks indicate significantly different values at *p* ≤ 0.05 (*) and *p* ≤ 0.001 (**).

**Table 1 ijms-23-12049-t001:** Hydrogen, acetic acid (AA), and lactic acid (LA) yields (mol/mol glucose) at 24 and 48 h. Letters indicate ANOVA significant differences between treatments, chemical products and bacterial species. Data are expressed as mean ± SD, *n* = 9.

		24 h				48 h		
	H_2_	AA	LA	LA/AA	H_2_	AA	LA	LA/AA
*Tnea* N_2_	2.60 ± 0.15 a	1.08 ± 0.02 b	0.21 ± 0.02 c	0.20 ± 0.02	2.40 ± 0.12 a	1.03 ± 0.03 b	0.26 ± 0.01 c	0.25 ± 0.01
*Tnea* CO_2_	2.04 ± 0.07 a	0.98 ± 0.03 b	0.36 ± 0.06 d	0.37 ± 0.05	2.41 ± 0.08 a	0.94 ± 0.06 b	0.45 ± 0.03 e	0.48 ± 0.02
*Trq7* N_2_	2.31 ± 0.04 a	1.13 ± 0.07 b	0.22 ± 0.01 c	0.19 ± 0.01	2.64 ± 0.04 a	1.09 ± 0.06 b	0.27 ± 0.01 c	0.25 ± 0.01
*Trq7* CO_2_	2.17 ± 0.26 a	0.95 ± 0.09 b	0.46 ± 0.05 e	0.48 ± 0.03	2.56 ± 0.26 a	0.96 ± 0.08 b	0.47 ± 0.03 e	0.49 ± 0.05
*Tmar* N_2_	2.42 ± 0.82 a	0.96 ± 0.01 b	0.12 ± 0.04 f	0.10 ± 0.06	2.67 ± 0.04 a	1.07 ± 0.11 b	0.07 ± 0.01 f	0.06 ± 0.00
*Tmar* CO_2_	2.60 ± 0.89 a	0.99 ± 0.20 b	0.11 ± 0.03 f	0.09 ± 0.00	2.33 ± 0.03 a	1.15 ± 0.12 b	0.06 ± 0.01 f	0.05 ± 0.00

**Table 2 ijms-23-12049-t002:** F-type and V-type ATPases locus tags in *Tnea*, *Trq7*, and *Tmar*.

Enzymes	*Tnea*	*Trq7*	*Tmar*
V-ATPase subunit A-atpA	CTN_RS04540	TRQ7_06035	Absent
V-ATPase subunit B-atpB	CTN_RS04545	TRQ7_06040	Absent
V-ATPase subunit D-atpD	CTN_RS04550	TRQ7_06045	Absent
V-ATPase subunit E-atpE	CTN_RS04535	TRQ7_06030	Absent
V-ATPase subunit F-atpF	CTN_RS04525	TRQ7_06020	Absent
V-ATPase subunit G-atpG	CTN_RS04530	TRQ7_06025	Absent
F-ATPase subunit A-atpB	CTN_RS04145	TRQ7_05640	TM_1616
F-ATPase subunit B-atpF	CTN_RS04155	TRQ7_05650	TM_1614
F-ATPase subunit C-atpE	CTN_RS04150	TRQ7_05645	TM_1615
F-ATPase subunit Delta-atpH	CTN_RS04160	TRQ7_05655	TM_1613
F-ATPase subunit Epsilon-atpC	CTN_RS04180	TRQ7_05675	TM_1609
F-ATPase subunit Gamma-atpG	CTN_RS04170	TRQ7_05665	TM_1611
F-ATPase subunit Alpha-atpA	CTN_RS04165	TRQ7_05660	TM_1612
F-ATPase subunit I-atpI	CTN_RS04140	TRQ7_05635	-
F-ATPase subunit Beta- atpD	CTN_RS04175	TRQ7_05670	TM_1610

## Data Availability

Not applicable.

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
