# Peer review of "Occurrence of Capnophilic Lactic Fermentation in the Hyperthermophilic Anaerobic Bacterium Thermotoga sp. Strain RQ7"

_ijms, 2022, doi:10.3390/ijms231912049_

Round 1

Reviewer 1 Report

Major comments

Figure 2. The graphics show the glucose depletion in 2 conditions: sparging N2 or CO2. Although the results are clear regarding the differences between both gases, I wonder if the authors have achieved the experiment with no sparging any gas. Could the sparging have any effect in the behavior of the strains?

Figure 4 and Table 1. According to the results, Tnea and Trq7 strains growing with CO2 sparging double the lactate production keeping the same acetate production. Considering the CLF pathway of Figure 1, the increase in lactate production could be explained by the displacement of PFOR reaction from acetyl-CoA to pyruvate (and then, to lactate). In this situation, it reasonable to expect a decrease in acetate production, because acetyl-CoA is been produced from acetate. However, the results show that acetate production keeps at the same level. How can be explained these data?

Figure 5. Regarding to the HYDa expression, there is a significant difference between N2 and CO2 condition in the case of Tnea and Trq7. However, attending the Figure 3, the H2 production was very similar in both conditions. Moreover, comparing HYD expression between Tnea and Trq7, we can see that Tnea shows double fold change (around 6) than Trq7 (around 3) in presence of CO2. However, in the Figure 3 we could observe that both strains produce similar levels of H2. These data could indicate that H2 production is not coming from HYDa. How do you explain these differences?

Minor comments

Line 37: change “Pseudethermotoga” by “Pseudothermotoga”.

Figure 2: in the X-axis, change “hour” by “time (h)”.

Figure 3: in the Y-axis, change “colture” by “culture”.

Figures 5 and 6: increase the font size in the X- and Y-axis.

Line 210: change “has” by “have”.

Supplementary: There are 2 Tables with the name “Table S2”. Change the first one by “Table S1”.

Line 305: remove the reference and keep the same previous format, with numbers.

Sometimes, the word milliliter is written in this way “ml” or in this way “mL”. Please, keep the same format along the paper.

Author Response

We thank the reviewer for hi/her valuable suggestions which help us to improve the manuscript.

Reviewer 2 Report

This manuscript describes the capnophilic fermentation ability of strain RQ as T. neapolitana and T. maritima as references.

Although the research is soundly performed, the text lacks clarity. The main focus seems to be the strain RQ. However, this is not understood from the title and abstract. The manuscript  should be extensively rewritten to increase readability.

Author Response

We thank the reviewer for his/her valuable suggestions which help us to improve the manuscript.

Round 2

Reviewer 1 Report

After the changes introduced by the authors, I think that the manuscript is ready to be published.

Reviewer 2 Report

The readability of the manuscript greatly improved.